# Single-cell lineage tracing in the mammary gland reveals stochastic clonal dispersion of stem/progenitor cell progeny

Felicity M. Davis[1,2,*], Bethan Lloyd-Lewis[1,*], Olivia B. Harris[1,3], Sarah Kozar[4], Douglas J. Winton[4], Leila Muresan[5] & Christine J. Watson[1,3]

The mammary gland undergoes cycles of growth and regeneration throughout reproductive life, a process that requires mammary stem cells (MaSCs). Whilst recent genetic fate-mapping studies using lineage-specific promoters have provided valuable insights into the mammary epithelial hierarchy, the true differentiation potential of adult MaSCs remains unclear. To address this, herein we utilize a stochastic genetic-labelling strategy to indelibly mark a single cell and its progeny *in situ*, combined with tissue clearing and 3D imaging. Using this approach, clones arising from a single parent cell could be visualized in their entirety. We reveal that clonal progeny contribute exclusively to either luminal or basal lineages and are distributed sporadically to branching ducts or alveoli. Quantitative analyses suggest that pools of unipotent stem/progenitor cells contribute to adult mammary gland development. Our results highlight the utility of tracing a single cell and reveal that progeny of a single proliferative MaSC/progenitor are dispersed throughout the epithelium.

[1] Department of Pathology, University of Cambridge, Cambridge CB2 1QP, UK. [2] School of Pharmacy, The University of Queensland, Brisbane 4072, Australia. [3] WellcomeTrust-Medical Research Council Cambridge Stem Cell Institute, University of Cambridge, Cambridge CB2 1QR, UK. [4] Cancer Research UK Cambridge Institute, University of Cambridge, Li Ka Shing Centre, Cambridge CB2 0RE, UK. [5] Cambridge Advanced Imaging Centre, University of Cambridge, Cambridge CB2 1QP, UK. * These authors contributed equally to this work. Correspondence and requests for materials should be addressed to C.J.W. (email: cjw53@cam.ac.uk).

The mammary epithelium is comprised of a highly branched, bilayered ductal tree with an inner layer of cytokeratin (K)8/18-expressing luminal cells and a surrounding layer of basal cells that typically express K5/14 and the contractile protein alpha-smooth muscle actin (SMA)[1]. As presumptive targets for transformation in breast cancer, the identity of adult mammary stem cells (MaSCs) and the origin of luminal and basal cell lineages have been the subject of intense investigation and debate[2]. Whilst results from transplantation assays in the mouse mammary gland point to the existence of bi/multipotent MaSCs that reside in the basal compartment[3–5], early genetic fate-mapping studies using lineage-specific promoters demonstrated that distinct unipotent MaSCs maintain the luminal and basal lineages postnatally under physiological conditions[6]. Subsequent lineage-tracing studies have provided further evidence in support of each model[7–10]. However, transplantation assays have been suggested to re-programme cells[6,8,10], and conventional lineage-tracing approaches have relied on prior assumptions regarding the specificity and consistency of expression of lineage markers. Moreover, these studies have induced labelling at levels significantly higher than clonal density, thus confounding analysis of daughter cells and their contribution to different lineages. Consequently, we have sought to resolve these complexities and determine unequivocally the potential of adult MaSCs, during puberty and pregnancy, by combining the use of mouse models that enable labelling of a single random cell and all of its progeny in situ with confocal three-dimensional (3D) imaging.

## Results

### Optical clearing and 3D imaging of the intact mammary gland.
To accurately determine the capacity of a single marked stem or progenitor cell and its progeny to contribute to the development of the branching mammary epithelial network in vivo, the entire ductal tree needs to be visualized at high spatial resolution. The utility of 3D imaging for fate-mapping studies has previously been demonstrated in stroma-divested mammary glands[7,9]. In this study we developed and refined methods for 3D imaging in the mammary gland, using techniques for optical tissue clearing to enable visualization of the mammary epithelium at single-cell resolution, without the need for enzymatic digestion or mechanical dissection.

Tissue clearing methods that have been developed are based on mitigating light scattering caused by cellular and extracellular structures with different refractive indices[11]. However, the utility of these protocols in the mammary gland remains largely unexplored. We determined that the SeeDB[12] and the CUBIC[13] tissue-clearing protocols provide superior optical clarity in mammary tissue (Fig. 1a). Moreover, by combining optical tissue clearing with wholemount immunostaining (Supplementary Fig. 1a) and algorithms to improve the signal-to-noise ratio of 3D image sequences[14] (Supplementary Fig. 1b), we were able to visualize the epithelial ductal tree to depths exceeding 400 μm, identifying K5-expressing basal cells and K8-expressing luminal cells at single-cell resolution (Fig. 1b, Supplementary Fig. 2 and Supplementary Movies 1 and 2). We could also image lactating mammary glands in 3D, highlighting the organization of basal cells by SMA immunostaining and luminal cells by their expression of E-cadherin and K8 (Fig. 1c and Supplementary Movie 3). We noted that K8 is non-uniformly expressed in luminal cells (Fig. 1b,c and Supplementary Movie 2) and identified two distinct subpopulations comprising K8$^{lo}$ and K8$^{hi}$ cells in situ, the latter co-staining with nuclear progesterone receptor (PR) (Fig. 1d and Supplementary Fig. 3). While these K8 subpopulations are present

in similar proportions in virgin ducts, K8$^{hi}$ cells are extremely sparse in lactational alveoli (Fig. 1c and Supplementary Fig. 3c). An association between high K8/18 expression and the functionally distinct CD24$^{hi}$/prominin-1$^{+}$/Sca1$^{+}$ hormone-sensing luminal population has previously been observed[15]. Moreover, lineage-tracing studies using K8-CreER$^{T2}$/Tomato-reporter mice have demonstrated preferential genetic labelling of CD24$^{hi}$/Sca1$^{+}$ luminal cells[16]. Collectively, these observations question the utility of the promoters of keratins and other presumed lineage-specific genes as suitable drivers of reporter proteins for lineage-tracing studies.

### A stochastic labelling strategy for single-cell lineage tracing.
To avoid prior assumptions regarding the expression profile of MaSCs and to track the fate of a single marked cell in the mammary epithelium, we utilized R26$^{[CA]30}$ reporter mice, which have previously been used to infer stem cell dynamics in the intestinal epithelium[17]. This model encompasses a dinucleotide repeat tract, [CA]$_{30}$, positioned downstream of the translational start site of an out-of-frame reporter gene (enhanced yellow fluorescent protein (EYFP) or modified β-glucosidase (SYNbglA)) inserted in the constitutively expressed Rosa26 locus (Fig. 2a). The inherent instability of microsatellite repeats can lead to spontaneous, random frame-shift mutations during DNA replication, which may place the reporter gene in-frame, thereby resulting in its expression. The advantages of this labelling approach are twofold: first, replication slippage is equally likely to occur in all cycling cells; and second, strand slippage is extremely rare[17], thus allowing all of the progeny of a single labelled cell to be identified with confidence.

### Clonal labelling patterns in the mouse mammary gland.
To determine the suitability of this model for single-cell lineage tracing in the mammary epithelium we examined clone abundance, size and distribution in R26$^{[CA]30SYNbglA}$ mice during pubertal development, when functionally active MaSCs are presumed to drive ductal elongation and branching morphogenesis[18,19]. These mice contain a modified β-glucosidase gene, which is thermostable and resistant to epigenetic silencing, downstream of the [CA]$_{30}$ tract (Fig. 2a), enabling macroscopic clonal analysis by wholemount histochemistry. Using this model, combined with CUBIC-based tissue clearing, regions of ducts containing variable numbers of β-glucosidase$^{+}$ cells interspersed with unlabelled cells could be visualized in situ (Fig. 2b–d and Supplementary Figs 4 and 5). As in the intestine, strand slippage was extremely rare in the mammary epithelium, with ~1.49 ± 0.92 total labelling events observed per gland (Supplementary Fig. 4) and, as such, the likelihood of clone convergence in this model is exceedingly low. We observed large contiguous clonal regions containing several hundred label-positive cells that spanned numerous branching ducts (Fig. 2b,c and Supplementary Fig. 5). These were considered to have arisen from a single MaSC or progenitor. Isolated regions that contained limited numbers of label-positive cells were also observed (Supplementary Fig. 4), most likely the result of strand slippage in more differentiated cells or in progenitors with restricted replicative potential (for example, Elf5-expressing luminal progenitors[7]). However, given the continual, albeit rare, genesis of labelled cells in this model, the possibility of recent strand slippage in a MaSC or highly proliferative progenitor could not be excluded. Label-positive regions were also detected after multiple pregnancy/involution cycles (Supplementary Fig. 6), indicating that some progeny may be long-lived.

Whilst strand slippage in a germ cell resulted in complete and uniform genetic labelling of ducts (Supplementary Fig. 7), clonal

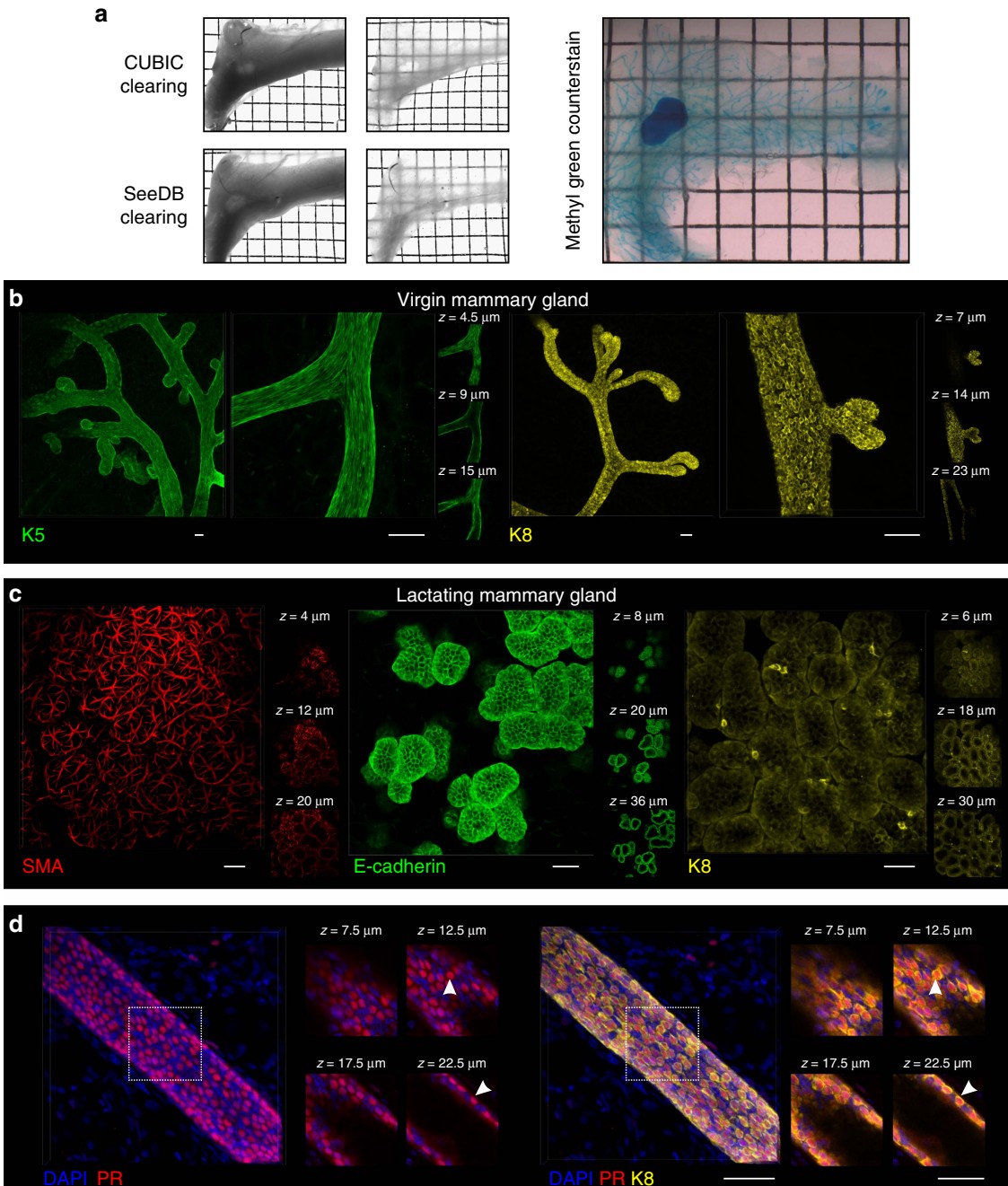

**Figure 1 | Optical tissue clearing and 3D imaging of the mammary gland. (a)** Transmission images of an entire virgin mammary gland before (left panels) and after (middle panels) tissue clearing using CUBIC or SeeDB, and a CUBIC-cleared mammary gland counterstained with methyl green to visualize the complete ductal network (right panel). Representative images from three mice. Grid width: 2 mm. **(b)** 3D imaging of K5 and K8 immunostaining of SeeDB-cleared virgin mammary tissue within its native stroma. K5 overview shows 1.18 mm ($xy$) of mammary gland ($z = 114\,\mu m$ imaging stack depth); K8 overview shows $834\,\mu m$ ($xy$) ($z = 114\,\mu m$). The depth ($z$) is relative to the first image in the image sequence, reached after passing through the mammary fat pad ($\sim350\,\mu m$). Scale bars, $50\,\mu m$. **(c)** Immunostaining for SMA ($xy = 579\,\mu m$; $z = 53\,\mu m$), E-cadherin ($xy = 467\,\mu m$; $z = 36\,\mu m$) and K8 ($xy = 399\,\mu m$; $z = 32\,\mu m$) in mammary glands from lactating mice. Scale bars, $50\,\mu m$. **(d)** Immunostaining shows populations of K8[hi] and K8[lo] luminal cells, with the K8[hi] cells costaining with nuclear PR (arrowhead) (representative images from three mice); scale bars, $50\,\mu m$.

expansion from a single MaSC/progenitor produced a stochastic labelling pattern, with β-glucosidase[+] cells intermixed randomly with unlabelled cells in branching ducts spanning over 8 mm in length (Fig. 2b–d and Supplementary Fig. 5). These clonal labelling patterns strongly suggest that a pool of active mammary stem/progenitor cells reside within each terminal end bud (TEB), the presumptive origin of MaSCs[20], and contribute to

the development of each major duct during puberty. The unequal distribution of labelled progeny between branching ducts (for example, Fig. 2b) is most likely due to the dilution of marked daughter cells with the progeny of unmarked MaSCs/progenitors during TEB bifurcation or secondary branching.

Labelled progeny arising from the expansion of a single β-glucosidase[+] cell had a luminal-like morphology (Fig. 2b and

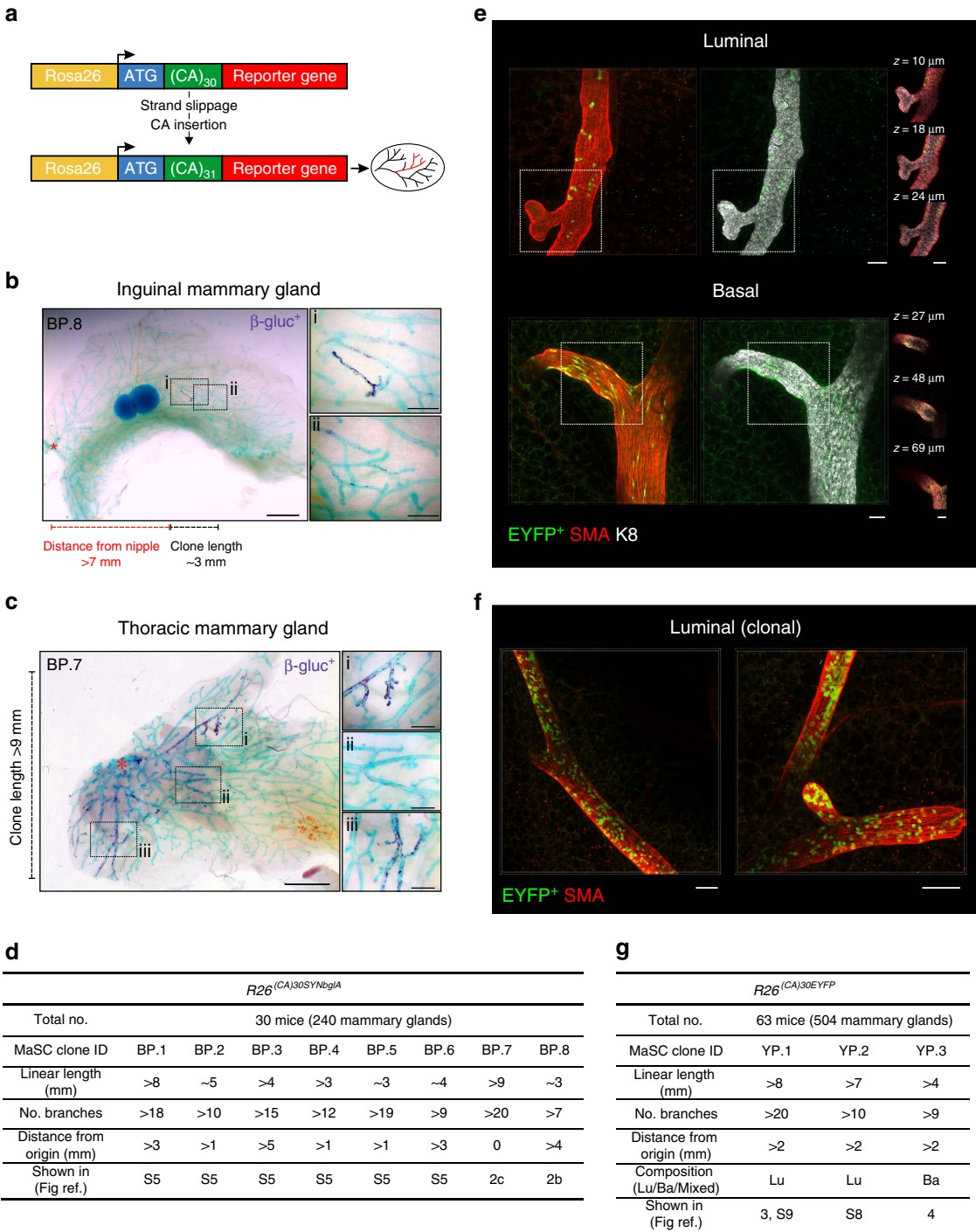

**Figure 2 | Single-cell lineage tracing in the virgin mammary gland.** (**a**) Schematic representation of the $R26^{[CA]30}$ mouse model. (**b,c**) Examples of two large clonally marked regions (BP.8 and BP.7) in mammary glands from $R26^{[CA]30SYNbglA}$ mice that were likely to have arisen from the labelling of a MaSC/progenitor (based on linear length and number of label-positive branches) (**d**). Dark purple staining is β-glucosidase$^+$ cells; mammary tissue was counterstained with methyl green. Annotations show the linear length of the clones and their distance from the nipple region (asterisk). Clone BP.7 originated in the nipple region and is likely to have been labelled very early in development. Scale bars, 2 mm (overview) and 0.5 mm (inset). (**d**) A summary of the eight clonally marked regions likely to have arisen from the labelling of a MaSC/progenitor, observed from the analysis of 30 $R26^{[CA]30SYNbglA}$ mice. (**e**) Examples of luminal (top panel) and basal (bottom panel) EYFP$^+$ cells from $R26^{[CA]30EYFP}$ mice representing over 25 label-positive regions. Scale bars, 50 μm. (**f**) A large clonally marked region containing many EYFP$^+$ cells. Labelled progeny spanned multiple ducts and exhibited a sporadic labelling pattern, intermixed with unlabelled cells. Scale bars, 100 μm. (**g**) A summary of three clonally marked regions presumed to have arisen from the labelling of a MaSC/progenitor, observed from the analysis of 63 $R26^{[CA]30EYFP}$ mice. Lu, luminal; Ba, basal.

Supplementary Fig. 5) in the majority of label-positive regions. Regions with basal-like β-glucosidase$^+$ cells were observed but were less common, possibly reflecting a smaller proportion of proliferating cells in this cellular compartment[21,22]. On one occasion we observed ducts that appeared, on the basis of morphology, to be comprised exclusively of β-glucosidase$^+$

luminal cells connected to ducts comprised exclusively of β-glucosidase$^+$ basal cells (Fig. 2c(i) and (iii) versus (ii)). The proximity of these diverging ducts suggests that labelled cells arose from a single, bipotent parent cell that gave rise to one luminal and one basal daughter. Given the expansive nature of this exceptional clone (BP.7, >9 mm in length), and the presence of labelled cells in the nipple region, it is likely that a bipotent MaSC was genetically marked very early in development of the gland, possibly during embryogenesis[6,23], and later gave rise to lineage-restricted progeny in the postnatal gland. This juxtaposition of presumptive luminal and basal ductal clones provides intriguing new insights into the likely fate of progeny of a bipotent embryonic MaSC that could only be revealed by single-cell labelling.

**Unipotent cells contribute extensively to ductal morphogenesis.** To examine more closely the clonal labelling patterns arising from adult MaSCs/progenitors, and to more definitively determine lineage on the basis of appropriate markers, we utilized the $R26^{[CA]30EYFP}$ mouse model, combined with SeeDB-based optical tissue clearing. Using this approach we were able to visualize and characterize progeny arising from a single fluorescently marked cell *in situ* with single-cell resolution (Fig. 2e,f). We note that despite the high degree of optical clarity achieved using this method, some regions deep within the mammary fat pad could not be visualized at single-cell resolution by confocal microscopy and thus a larger number of mice were required for analysis in this model. Immunolabelling for markers of basal (SMA) and luminal (K8) lineages confirmed that the majority of labelled clones were luminal, with few basal clones observed (Fig. 2e). Only one large EYFP$^+$ basal clone, spanning over nine branches, which could have arisen from a stem cell, was observed (Fig. 2g). Clonal expansion of a single EYFP$^+$ luminal cell produced a mosaic labelling pattern identical to those observed in the $R26^{[CA]30SYNbglA}$ model (Fig. 2f), confirming that more than one luminal MaSC/progenitor contributes to the elongation of each major duct during puberty. Since the timing of the slippage event cannot be determined, we measured both the length of each clone and the distance from the nipple region where labelled cells are first observed (Fig. 2g and Supplementary Fig. 8, clone YP.2). If the labelled cell of origin is more than 1 mm from the nipple region, our assumption is that slippage has occurred in a stem or progenitor cell postnatally[20]. All such large clonal regions were lineage-restricted and we did not detect luminal and basal EYFP$^+$ cells intermingled within the same duct (Fig. 2g). These data support our observations with the $R26^{[CA]30SYNbglA}$ mice and provide further compelling evidence that unipotent MaSCs/progenitors contribute extensively to ductal morphogenesis[6]. However, due to the requirement for proliferation to label and trace stem and progenitor cells and the aforementioned issues associated with deep imaging, we cannot rule out the possibility that rare quiescent bipotent MaSCs, not detected in the $R26^{[CA]30EYFP}$ model, may exist.

To confirm the lineage restriction of adult MaSCs/progenitors, and quantify their contribution to ductal morphogenesis, we analysed a large clonal region (Fig. 3a–e and Supplementary Fig. 9) using imaging algorithms for the volumetric segmentation of mammary ducts and the subsequent characterization of all ductal EYFP$^+$ cells (Methods and Fig. 3c). This clonal region was more than 8 mm in length and comprised over 20 branching ducts (Figs 2g and 3a and Supplementary Fig. 9). Since the clone originated more than 2 mm from the nipple region, this suggests that strand slippage most likely occurred postnatally. All EYFP$^+$ cells examined by 3D analysis expressed the luminal marker K8 (Fig. 3b), and encompassed both K8$^{hi}$ and K8$^{lo}$ subpopulations

(Fig. 3c and Supplementary Movie 4), with a modest but significant overrepresentation of the EYFP label in K8$^{hi}$ cells (Kolmogorov–Smirnov test at $P < 0.05$) (Supplementary Fig. 10a). The lineage restriction of this clone was also confirmed by histological analysis of sectioned tissue over depths of 300 µm (Fig. 3d and Supplementary Fig. 10b). To determine the potential contribution of an active MaSC/progenitor to each duct, a volumetric ratio of EYFP$^+$ cells with respect to the total cellular volume was computed, revealing that the parent cell contributed on average $4.7 \pm 1.7\%$ of the total cellular volume in this region of the clone (Fig. 3e). A similar analysis was performed for a basal clone (Fig. 4a–c), where it was determined that the parent basal MaSC/progenitor contributed on average $5.8 \pm 3.2\%$ of the total cellular volume (Fig. 4d). Whilst these numbers may reflect the differential proliferative and competitive behaviours of stem/progenitor cells and their progeny, in addition to their random distribution with branching, these data indicate that there may be at least 20 luminal and 15 basal lineage-restricted stem/progenitor cells in each major duct driving mammary gland morphogenesis during puberty.

**Neutral lineage tracing using a multicolour reporter.** Although the $R26^{[CA]30SYNbglA}$ and $R26^{[CA]30EYFP}$ models have provided unprecedented insights into the contribution of a single stem/progenitor cell to mammary gland development, we sought to confirm the labelling pattern during pubertal development with a different neutral approach that also allows the timing of the labelling event to be controlled and is not dependent on a cell being in cycle at the time of labelling. To achieve this, we utilized the Confetti multicolour reporter mouse[24], combined with wholemount immunostaining and 3D imaging. We generated mice that were hemizygous for both R26-Confetti (ref. 24) and R26-CreERT2 (ref. 25; Fig. 5a) and administered a single low-dose of tamoxifen to 4-week-old mice followed by a 3-week chase to trace the progeny of cells labelled at the onset of puberty (Supplementary Fig. 11a). This resulted in low-frequency[26] multicolour labelling in the mammary epithelium, allowing us to distinguish individual clones. To identify luminal and basal cells, wholemount immunostaining was performed using a lineage-specific marker (either K8 or SMA, respectively) and tissues were counterstained with 4′,6-diamidino-2-phenylindole (DAPI) to localize the mammary epithelium. Cyan fluorescent protein (CFP)-expressing clones were under-represented and were therefore not analysed. The explanation for this is not entirely clear but may relate to the poor penetration of short-wavelength light through thick specimens and the fine membranous localization of the CFP reporter protein. Using this approach, we were able to visualize luminal and basal lineage-restricted GFP$^+$, YFP$^+$ and RFP$^+$ clones (Fig. 5b,c and Supplementary Fig. 11b,c). Notably, labelling patterns in the confetti mice were similar to those in the $R26^{CA30}$ models, validating the latter approach.

**Unipotent cells contribute extensively to alveologenesis.** Finally, to investigate the contribution of a single adult MaSC/progenitor to the formation of lobuloalveolar structures during pregnancy, we analysed clonal labelling patterns in lactating $R26^{[CA]30EYFP}$ mice (Fig. 6 and Supplementary Figs 12 and 13). We noted an increased number of clonal regions in tissue from lactating mice compared with puberty (probably as a consequence of the higher levels of proliferation), and a striking variety of patterns. The unequal distribution of EYFP$^+$ cells between lobuloalveolar units could suggest that an alveolar stem cell niche is situated close to the branch point of the subtending ducts (Fig. 6a). Competition for niche occupancy may dictate the dispersal of labelled and

unlabelled daughter cells between adjacent lobules, with further competitive interactions between their respective progeny possibly determining labelling outcomes within each alveolus[27]. Indeed, alveoli that were comprised almost entirely of label-positive luminal cells were occasionally observed (Fig. 6b),

implying that the descendants of any unmarked luminal stem/progenitor cell had been outcompeted. We also observed an interesting pattern where many alveoli within a lobule contained only a single EYFP$^+$ cell (Fig. 6c) suggesting the possibility that the original labelled cell is restricted to a lineage that constitutes a

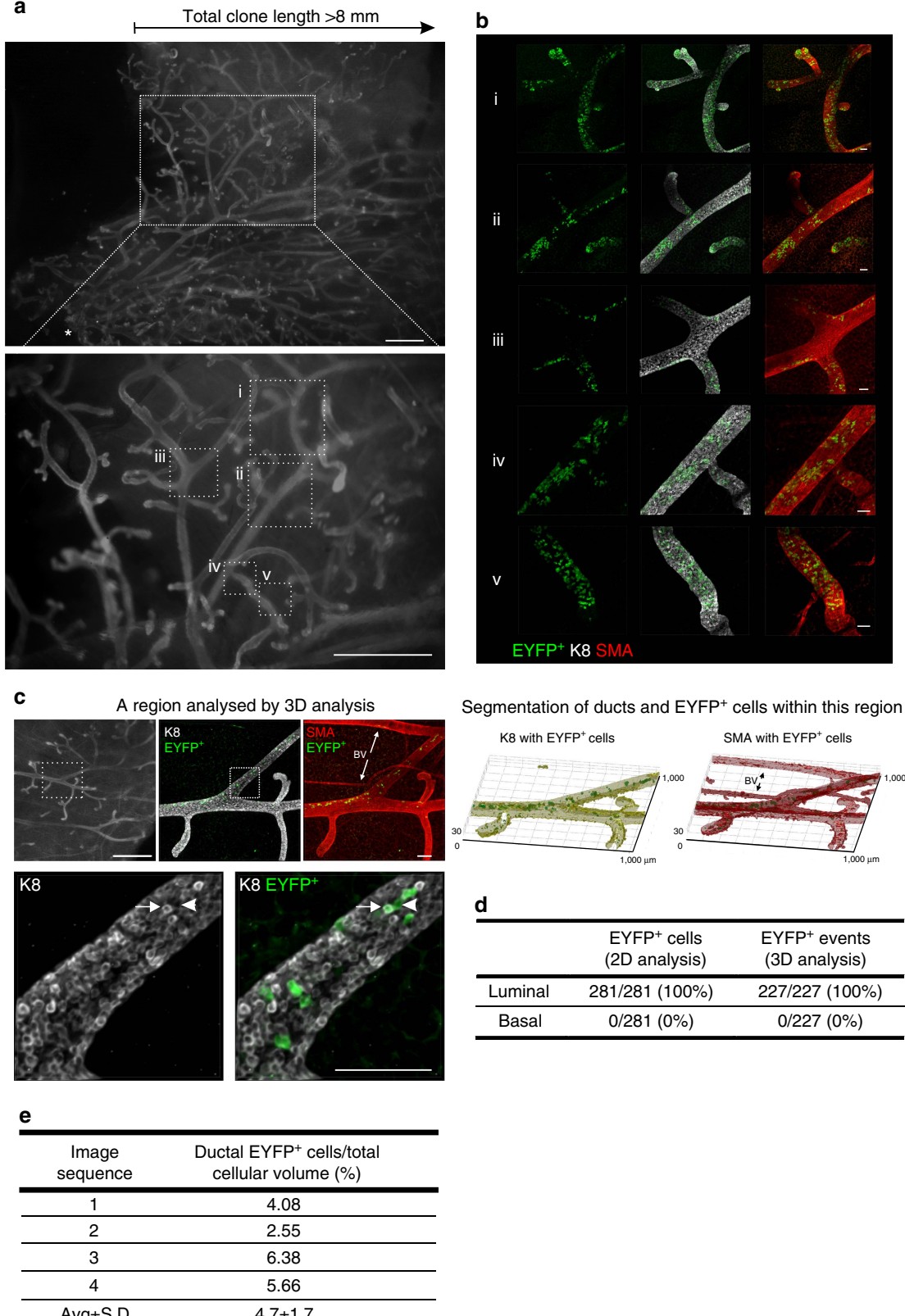

| | EYFP$^+$ cells (2D analysis) | EYFP$^+$ events (3D analysis) |
|---|---|---|
| Luminal | 281/281 (100%) | 227/227 (100%) |
| Basal | 0/281 (0%) | 0/227 (0%) |

| Image sequence | Ductal EYFP$^+$ cells/total cellular volume (%) |
|---|---|
| 1 | 4.08 |
| 2 | 2.55 |
| 3 | 6.38 |
| 4 | 5.66 |
| Avg±S.D | 4.7±1.7 |

minor, although possibly important, component of each alveolus. This has not been observed before and the identity and function of these cells is unclear. Similar to the pubertal epithelium, all large clonally marked regions were lineage-restricted, with separate luminal and basal clones observed during lactation (Fig. 6d and Supplementary Fig. 13). Of note, luminal cells contributed to both the K8[hi] hormone-sensing lineage, as well as the more abundant K8[lo] alveolar lineage (Supplementary Fig. 13, clone YL.1). In all cases, EYFP[+] progeny was intermixed with unlabelled cells in a polyclonal pattern that spanned numerous lobuloalveolar structures (Fig. 6d). Quantification of labelling patterns in these large clonally marked regions, some comprising over 100 alveoli (Fig. 6e), revealed that the majority of alveoli were comprised of both EYFP[+] cells and unlabelled cells of a single lineage (Fig. 6f), demonstrating that most alveoli are derived from at least two lineage-restricted stem/progenitor cells. These data contrast with previous studies suggesting that alveoli can be comprised of the progeny of a single cell[7,8]. The reason for this discrepancy is not clear but possibly reflects the uncertainties in analysing progeny of co-incident clones. Alternatively, stochastic stem cell fate could result in neutral drift and elimination of other stem cells and their progeny[28]. These insights into alveolar stem cell biology reflect the power of in vivo lineage tracing at clonal density.

## Discussion

The existence of MaSCs was demonstrated over 50 years ago[29]. More recently, the identity and potential of these cells has come under intense scrutiny, yet a number of uncertainties remain. Prime amongst these is whether MaSCs in the adult are unipotent or bipotent[6–9]. Although this may appear to be a relatively straightforward question to address, current experimental approaches have not provided an unequivocal answer. This is primarily a consequence of their dependence on presumed lineage-restricted promoters to drive reporter gene expression in a significant proportion of MaSCs for population-based fate tracking. Using this approach the probability of two or more clones arising in the same region is high, confounding their analysis. Thus, promiscuous labelling and subsequent expansion of even a single lineage-restricted MaSC could resemble clonal expansion of a bipotent MaSC[26]. In this context it is important to note that the expression of K14 and K18 is differentially regulated in the pre-pubertal mammary gland with some luminal cells unexpectedly expressing K14 (refs 30,31). Similar difficulties arise with the K8 promoter that is expressed at low levels in a subset of cells, leading to the disparate labelling of luminal cell populations in fate-mapping studies[16]. Thus, lineage tracing with these promoters is not definitive for the assessment of potency. We therefore adopted two agnostic fate-mapping strategies to avoid these confounding issues and this has resulted in a number of unanticipated observations and valuable insights that could only have been revealed by this stochastic single-cell-labelling approach.

Our first intriguing observation was the random distribution of labelled progeny of a single cell to multiple ducts (depicted schematically in Fig. 7a). Some regions had a high density of labelled cells while others had a much lower density, indicating the presence of multiple lineage-restricted stem cells and the admixing of their progeny. Imaging of entire mammary glands also revealed that all labelled ducts were connected, suggesting bifurcation and branching from a TEB in which the labelled cell presumably arose during puberty. We investigated the nature of cells within large clones using 3D imaging algorithms, and revealed that all labelled cells within these regions were lineage restricted. Furthermore, based on a volumetric analysis and the assumption that all MaSCs/progenitors have the same capacity to contribute to ductal outgrowth, we estimate that at least 20 luminal and 15 basal MaSCs/progenitors contribute to the growth of a major duct. By extension, this would equate to a few hundred unipotent luminal and basal MaSCs/progenitors per gland, which drive ductal morphogenesis during puberty. Our lactation data reveal the unexpected presence of different subpopulations of alveolar cells, including one type that contributes only a single cell to most alveoli in a lobuloalveolar cluster. We suggest that, as alveolar expansion during pregnancy is critically important, and may occur several times in a lifetime, a pool of alveolar stem or committed progenitor cells is required. The variable contribution of cells to individual alveoli (depicted schematically in Fig. 7b), with 100% contribution being rare, could reflect prior commitment to specific lineages or competition for the stem/progenitor cell niche[32,33].

We posit that the MaSCs/progenitors that generate the ductal network during puberty are distinct from those that have a more homeostatic function in the adult, the latter possibly arising from bipotent embryonic MaSCs that may persist after birth and remain quiescent[30]. Indeed, these cells would not have been labelled by our approach. Nevertheless, our data are consistent with unipotent mammary stem/progenitor cells being primarily present in the TEBs during puberty, where they proliferate and move towards the subtending duct as it elongates. These TEB-resident MaSCs/progenitors would be lost when the TEBs regress at the completion of puberty. However, slow-cycling unipotent MaSCs/progenitors may be deposited throughout the ductal network[34,35] and could later be recruited in response to pregnancy hormones to generate alveoli.

Our work has illuminated the capacity of a single cell in the adult mammary gland to contribute to mammary gland development. A complete resolution of the mammary stem cell hierarchy controversy will require the ability to label a single cell at a defined moment and follow its fate over time. Although fraught with difficulties, prospective isolation and transcriptome analysis of single MaSCs will be an aim for the future.

**Figure 3 | 3D analysis of a clone arising from a single labelled luminal presumptive MaSC.** (**a**) Wholemount fluorescence images (K8 immunofluorescence) of the mammary ductal network demarcating the linear length of the clone and one region (magnified views, i–v) that was imaged at high cellular resolution by confocal microscopy in **b**. Further regions from this clone are shown in Supplementary Fig. 9. Asterisk shows the location of the nipple. Scale bar, 1 mm (wholemount) and 50 μm (confocal). (**c**) Images of a clonally marked region that was analysed by 3D image analysis. Digital segmentation of EYFP[+] cells within the luminal (K8-expressing) and basal (SMA-expressing) compartments is shown. Original 3D images show that progeny from a single luminal MaSC/progenitor included both K8[hi] (arrow) and K8[lo] (arrowhead) cells. BV, blood vessel. Scale bars, 1 mm (wholemount) and 100 μm (confocal). (**d**) Tabulated results of 3D and two-dimensional (2D) clonal analyses. For 3D analysis, all segmented ductal EYFP[+] cells were classified as luminal based on the proportion of K8 versus SMA signal (n = 227 cells from 4 image sequences). For 2D analysis, cells were classified by manual scoring of histological sections (n = 281 cells from 10 sections spanning 300 μm depth). This clone (YP.1) is one of three clones likely to have arisen from the labelling of a MaSC (based on linear length and number of label-positive branches), identified from the analysis of over 500 mammary glands from 63 hemizygous R26[CA30EYFP] pubertal mice. (**e**) Tabulated results of the computed volumetric ratio of EYFP[+] cells with respect to cellular volume for each of the four regions analysed.

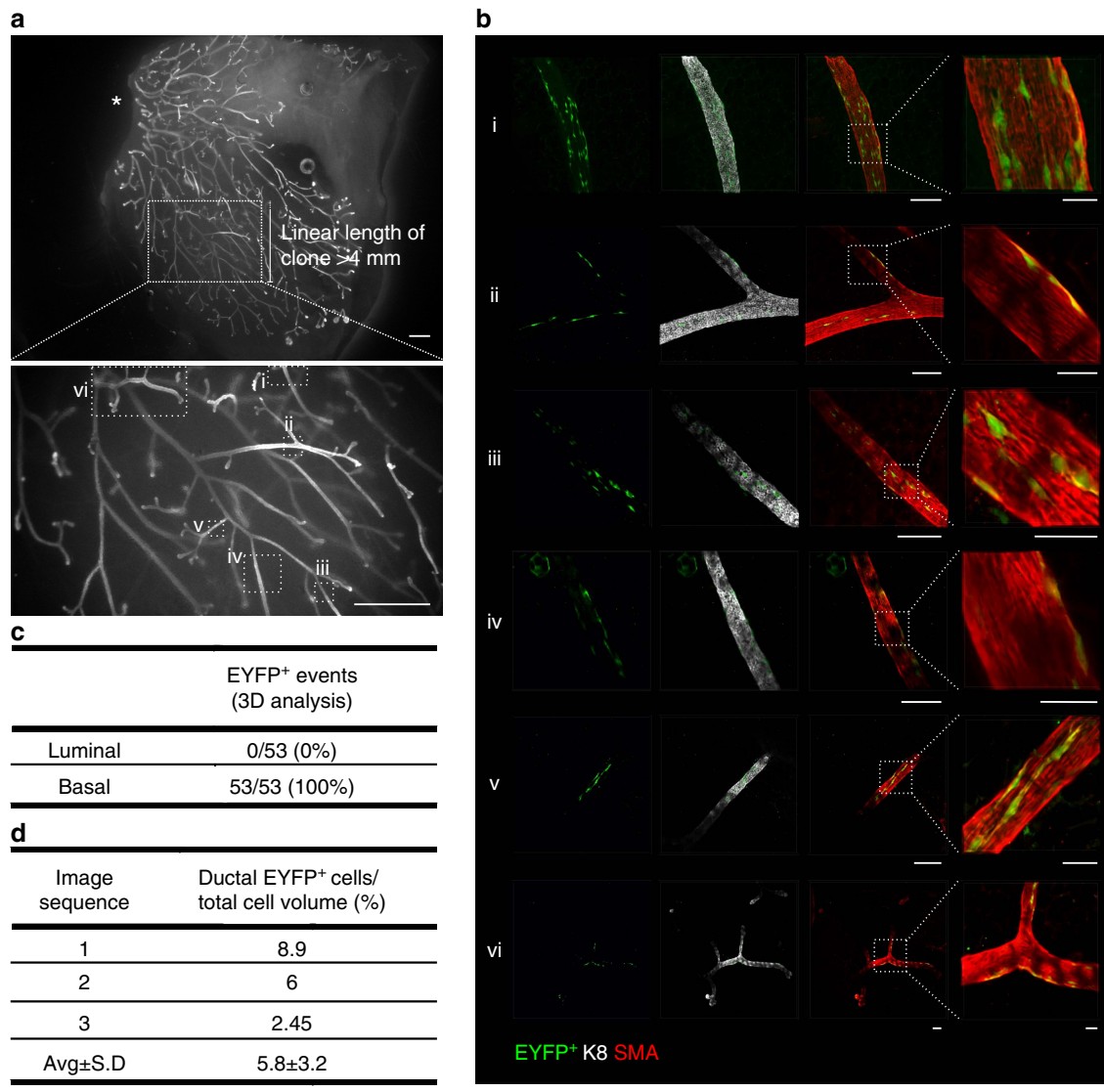

| | EYFP+ events (3D analysis) |
|---|---|
| Luminal | 0/53 (0%) |
| Basal | 53/53 (100%) |

| Image sequence | Ductal EYFP+ cells/ total cell volume (%) |
|---|---|
| 1 | 8.9 |
| 2 | 6 |
| 3 | 2.45 |
| Avg±S.D | 5.8±3.2 |

EYFP+ K8 SMA

**Figure 4 | 3D analysis of a clone arising from a single labelled basal presumptive MaSC.** (**a**) Wholemount fluorescence images (K8 immuno-fluorescence) of the mammary ductal network, mapping regions (i–vi) that were imaged using a confocal microscope in **b**. All EYFP+ cells in this clone (YP.3) were basal. A distinct clone (separated by >1mm) was identified in this tissue piece and contained only luminal cells (shown in Supplementary Fig. 8). Luminal and basal EYFP+ cells were not imaged in the same branches and clones were never intermixed. Asterisk shows the nipple and origin of the ductal network. Scale bar, 1mm (wholemount) and 100 μm (confocal, overview) or 30 μm (confocal, magnified view). (**c**) Tabulated results of the 3D lineage analysis (n = 53 cells from 3 image sequences). (**d**) Tabulated results of the computed volumetric ratio of EYFP+ cells with respect to total basal cellular volume for each of the three regions analysed.

## Methods

**Animal models.** *R26[CA]30SYNbglA* and *R26[CA]30EYFP* mice (on a C57Bl/6J background)[17] were provided by Prof. Douglas Winton (Cancer Research UK Cambridge Institute). Female virgin mice were killed by dislocation of the neck or terminal anaesthesia at 7 weeks of age for all studies in puberty. For single-cell lineage tracing in lactating mice, female *R26[CA]30* mice were mated with C57Bl/6J male studs and tissue was collected between lactation days 2–4. For analysis of multi-parous mice, female *R26[CA30]SYNbglA* mice were mated with C57Bl/6J male studs (for 3 pregnancy/involution cycles), and allowed to naturally litter and wean their pups. The final wean was followed by an 8- to 9-week interval before mammary tissue was collected. All quantitative analyses were performed on mice that were hemizygous for *R26[CA]30SYNbglA* or *R26[CA]30EYFP*. Mice that were hemizygous for both *R26-Confetti* (ref. 24) and *R26-CreERT2* (ref. 25) (*R26-Confetti;R26-CreERT2*) were generated by mating homozygous mice. These mice can theoretically produce cells that express either membrane-bound cyan, nuclear green, cytoplasmic yellow or cytoplasmic red fluorescent proteins following tamoxifen administration. Eight mammary glands (excluding the first (cervical) pair) were dissected and analysed for each mouse. Mammary glands were excised and fixed in 10% neutral buffered formalin (NBF) for 9 h at room temperature. Animals were housed in individually ventilated cages under a 12:12 h light–dark

cycle, with water and food available *ad libitum*. All animal experimentation was carried out in accordance with the Animal (Scientific Procedures) Act 1986, the European Union Directive 86/609 and with local ethics committee approval. No statistical method was used to predetermine sample size.

**Induction of lineage tracing.** Lineage tracing was induced at 28 days for studies in puberty in *R26-Confetti;R26-CreERT2* mice. A single intraperitoneal injection of tamoxifen (1 mg) in sunflower oil was administered and tissues were collected after a 2-day chase to determine initial labelling or after 3 weeks for pubertal lineage-tracing studies.

**Reagents.** The following reagents were purchased from Sigma Aldrich: NBF; urea; *N,N,N′,N′*-tetrakis(2-hydroxypropyl)ethylenediamine; 2,2′,2″-nitrilotriethanol; fructose; α-thioglycerol; DAPI dilactate; 3,3′-diaminobenzidine tetrahydrochloride (DAB); and tamoxifen. Sucrose was purchased from Fisher Scientific. Triton X-100 was purchased from VWR International. The following primary antibodies were used for immunostaining: rabbit anti-K5 (Covance, PRB160P, 1:100); rat anti-K8 (Developmental Studies Hybridoma Bank, TROMA-I, 1:50); rabbit anti-SMA (Abcam, ab5694, 1:300); mouse anti-SMA (Abcam, ab7817, 1;200); rabbit

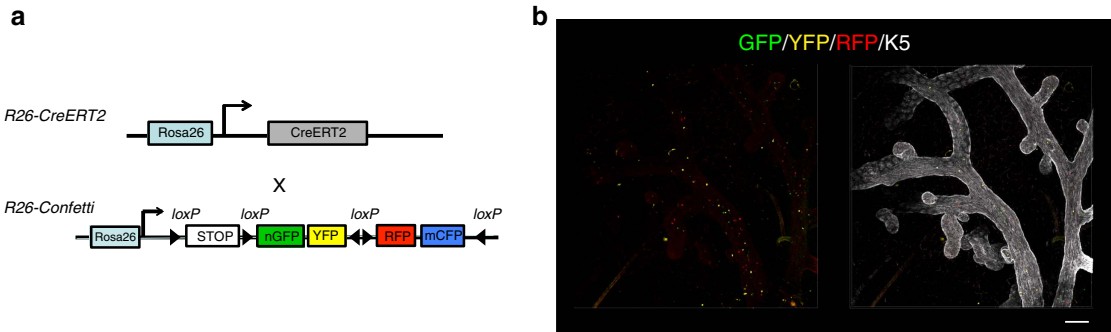

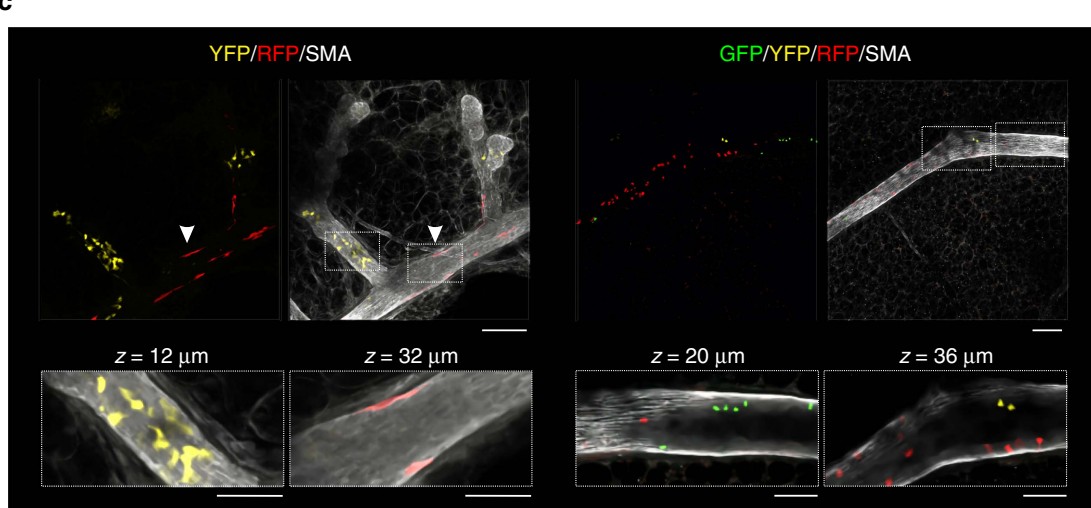

**Figure 5 | Clonal labelling patterns observed in *R26-Confetti;R26-CreERT2* pubertal mice.** (**a**) Schematic representation of the *R26-Confetti;R26-CreERT2* mouse model. (**b**) Initial labelling level observed 2 days after the administration of a single, low-dose of tamoxifen (1 mg intraperitoneal (i.p.)) to 4-week-old mice. Scale bar, 100 μm. (**c**) Labelling patterns observed in *R26-Confetti;R26-CreERT2* pubertal mice confirm the results from the *R26^[CA]30^* model. Labelling was induced by the administration of a single, low-dose of tamoxifen (1 mg i.p.) to 4-week-old mice and mammary glands were collected after a 3-week chase. Left panel shows a region containing YFP⁺ luminal cells and RFP⁺ basal cells (arrowhead) populating three branches, interspersed with unlabelled cells. Right panel shows a region containing GFP⁺, YFP⁺ and RFP⁺ luminal cells in a single branch. Scale bar, 100 μm (overview) and 50 μm (inset). Images are representative of three mice. Additional images are shown in Supplementary Fig. 11.

anti-E-cadherin (Cell Signaling, 3,195, 1:50); rabbit anti-PR (DAKO, A0098, 1:50); and chicken anti-GFP (Abcam, ab13970, 1:2,000). The following Alexa Fluor-conjugated secondary antibodies were purchased from Life Technologies and used 1:500: goat anti-mouse 488 (A11001); goat anti-mouse 647 (A21237); goat anti-rat Cy3 (A10522); goat anti-rat 488 (A11006); goat anti-rabbit 488 (A11008); goat anti-rabbit 647 (A21245); goat anti-chicken 488 (A11039); and goat anti-chicken 568 (A11041).

**Optical tissue clearing and wholemount immunostaining.** Mammary tissue was dissected and cut into large pieces (∼15 × 15 × 2 mm) for immunostaining and clearing. CUBIC-based tissue clearing was performed[13], with minor modifications, for visualization of clones from *R26^[CA]30SYNbglA^* mice. CUBIC Reagent 1 was prepared as a mixture of urea (25% w/w), N,N,N′,N′-tetrakis(2-hydroxypropyl) ethylenediamine (25% w/w) and Triton X-100 (15% w/w) in distilled water. CUBIC Reagent 2 was prepared using sucrose (44% w/w), urea (22% w/w), 2,2′, 2″-nitrilotriethanol (9% w/w) and Triton X-100 (0.1% w/w) in distilled water. Tissues were immersed in CUBIC Reagent 1 at 37 °C for 3 days. Mammary glands were counterstained for 1.5 h in methyl green (0.5%), washed and de-stained in acid alcohol. Tissues were immersed in CUBIC Reagent 2 at 37 °C for 1–2 days until transparent and imaged using a Leica MZ75 dissecting microscope.

SeeDB-based tissue clearing[12] was combined with wholemount immunolabelling for visualization of fluorescent clones from *R26^[CA]30EYFP^* and *R26-Confetti;R26-CreERT2* mice. Mammary tissue was blocked and permeabilized overnight at 4 °C in PBS with Triton X-100 (1%) and bovine serum albumin (10%). Primary antibodies were diluted in blocking buffer and incubated at 4 °C for 4 days with gentle rocking. Tissue was washed and incubated with secondary antibodies for 2 days at 4 °C before further washing and incubation with DAPI (10 μM) for 1–2 h. Samples were serially incubated for 8–16 h at room temperature

in 2–3 ml of 20, 40, 60 and 80% (w/v) fructose in distilled water, and subsequently transferred to 100% (w/v) fructose solution (24 h) and 115% (w/v) fructose solution for 24 h or until imaged. All fructose solutions contained α-thioglycerol (0.5%) to inhibit the Maillard reaction[12] and incubations were performed with gentle agitation.

**Detection of β-glucosidase expression.** For detection of modified β-glucosidase expression[36,37], mammary glands were excised and fixed for 4 h at room temperature in 10% NBF. Endogenous β-glucosidase activity was heat inactivated for 15 min at 65 °C in phosphate-buffered saline (PBS). Wholemount mammary glands were incubated for 48 h at 50 °C in a solution containing 1 part Solution A (5-bromo-6-chloro-3-indolyl-β-D-glucopyranoside (1%) in dimethyl sulfoxide) and 25 parts Solution B (magnesium chloride (0.02% w/v), potassium ferricyanide (0.096% w/v) and potassium ferrocyanide (0.13% w/v) in PBS), with substrate replenishment after 24 h. Mammary glands were post-fixed in 10% NBF overnight at 4 °C. Tissue clearing was performed using the CUBIC clearing protocol. For all animals, ∼5 cm of the small intestine distal from the stomach was excised and used as a reaction control for the detection of β-glucosidase expression.

**Histology and two-dimensional immunostaining.** For histological analysis of tissue from *R26^[CA]30EYFP^* mice, SeeDB-based optical tissue clearing was reversed by overnight incubation in PBS at 4 °C. Standard protocols for paraffin processing and embedding using alcohol and xylene were used. Paraffin-embedded sections (6 μm) were de-waxed in xylene, and antigen retrieval was performed by boiling in tri-sodium citrate buffer (10 mM, pH 6.0), for 11 min. Sections were blocked in goat serum (5%) in PBS supplemented with 0.05% Triton X-100 PBS for 1 h at room temperature. Sections were incubated with primary antibodies overnight at 4 °C. Primary antibodies used were as follows: rat anti-K8 (1:200); chicken

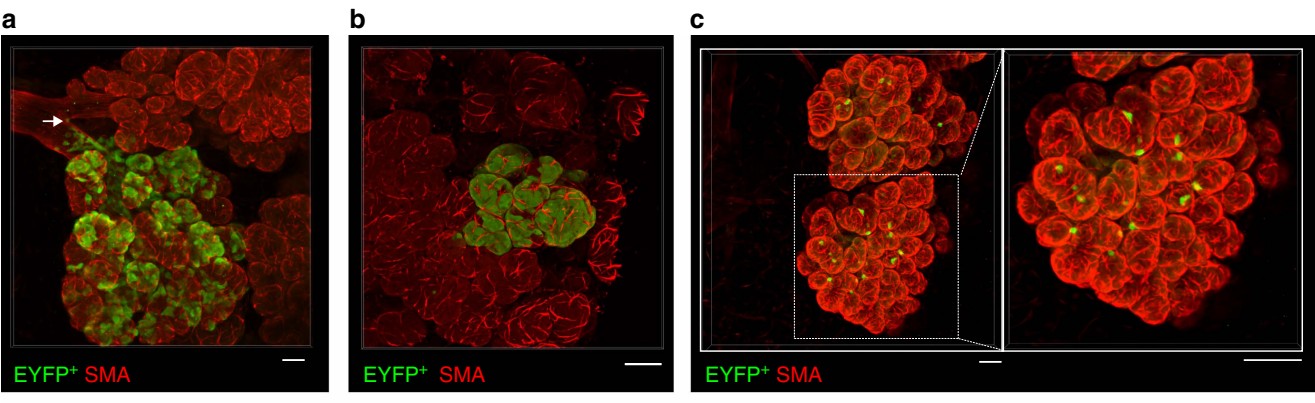

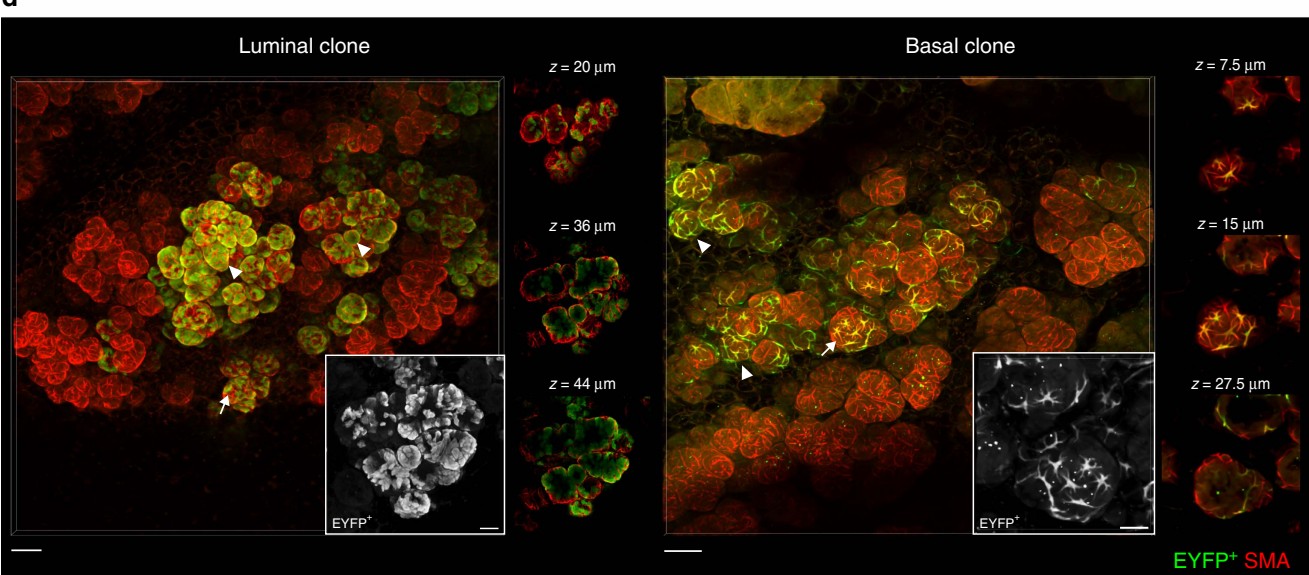

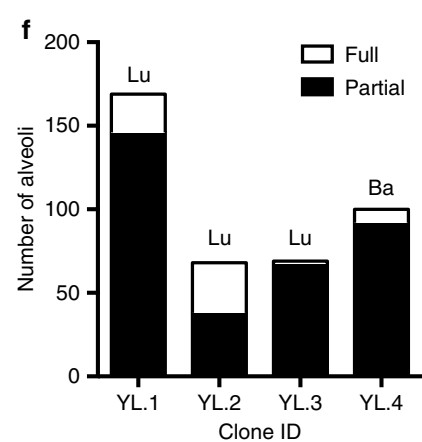

**e**

| | *R26*$^{(CA)30EYFP}$ | | | |
|---|---|---|---|---|
| Total no. | 10 mice (80 mammary glands) | | | |
| MaSC clone ID | YL.1 | YL.2 | YL.3 | YL.4 |
| Linear length (mm) | >2 | >1 | >1 | >2 |
| No. alveoli | >169 | >68 | >69 | >100 |
| Composition | Lu | Lu | Lu | Ba |
| Shown in (Fig ref.) | 6, S13 | S13 | S13 | 6, S13 |

**Figure 6 | Contribution of a single MaSC/progenitor to alveologenesis.** (**a**) An image showing the uneven contribution of a single labelled ductal cell to different lobuloalveolar structures; arrow indicates the presumptive EYFP$^+$ cell of origin at the ductal branch point. Scale bar, 50 µm. (**b**) A rare instance of progeny from a single luminal EYFP$^+$ cell contributing almost entirely to the luminal lineage of 2–4 alveoli within a single lobule. Scale bar, 50 µm. (**c**) An example of a single labelled EYFP$^+$ luminal cell that contributed one EYFP$^+$ luminal daughter cell to multiple alveoli in a lobule. Scale bar, 100 µm. (**d**) 3D images revealing the extensive contribution of a single luminal (left, clone YL.1) and basal (right, clone YL.4) EYFP$^+$ cell to the lobuloalveolar network in independent lactating mammary glands. Labelled alveoli were mostly populated by both lineage-restricted EYFP$^+$ and unlabelled cells (arrow), with occasional alveoli observed that were fully populated by EYFP$^+$ cells of a single lineage (arrowhead). Scale bar, 100 µm (overview) and 40 µm (inset). (**e**) A summary of the four large clonally marked regions observed from the analysis of 10 *R26*$^{[CA]30EYFP}$ mice during lactation. (**f**) The number of alveoli that were fully populated by EYFP$^+$ cells of a single lineage (full) or populated by both EYFP$^+$ and unlabelled cells of a single lineage (partial). Lu, luminal; Ba, basal.

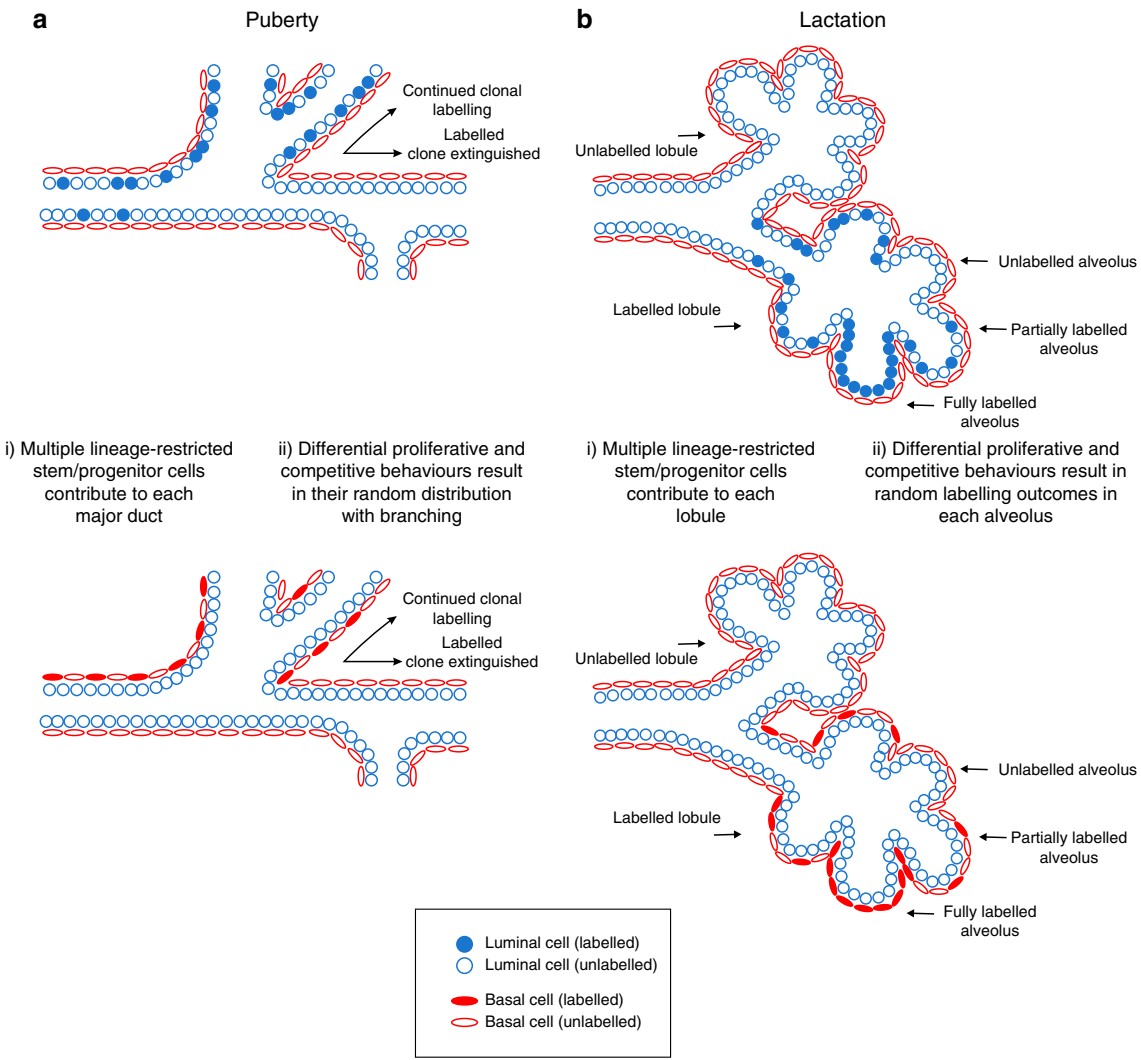

**Figure 7 | Schematic model.** Schematic model of the most common labelling pattern arising from the genetic labelling of a single lineage-restricted MaSC/progenitor to (**a**) ductal morphogenesis and (**b**) alveologenesis, identified in *R26[CA]30* mice and confirmed using the *R26-Confetti;R26-CreERT2* model (puberty).

anti-GFP (1:2,000); and rabbit anti-SMA (1:200). Secondary antibodies were diluted 1:500. Nuclei were counterstained with DAPI (5 μM).

**Confocal microscopy and image analysis.** Images of wholemount mammary glands were acquired using a Leica TCS SP8 inverted confocal microscope with 10 × /0.4 or 20 × /0.75 HC PL APO objective lenses. Laser power, line averaging and step increment were adjusted manually to give optimal fluorescence intensity for each fluorophore with minimal photobleaching. Imaging depths were recorded from the top of the epithelial structure being imaged. However, an actual imaging depth of ∼350 μm through the native fat pad was typically required before reaching the mammary ductal tree. Thus, imaging depths were routinely 350–500 μm through the tissue. An XZ projection illustrating the lower axial resolution versus the lateral resolution is shown in Supplementary Fig. 14a. The reduced resolution had little effect on the 3D image analyses, with EYFP⁺ cells able to be discriminated on the anterior and posterior surfaces of most ducts. CFP-expressing clones were rarely observed in *R26-Confetti;R26-CreERT2* mice, possibly related to the brightness of the fluorescent protein, poor penetration of short-wavelength light through the lipid-rich mammary fat pad, the fine membranous localization of the CFP reporter protein, as well as the available laser lines on the confocal microscope. Image reconstructions were generated using Imaris image management software (v8.0, Bitplane). Denoizing of 3D image sequences was performed in MATLAB[14].

Analyses of 3D image stacks, selected on the basis of their resolution and compatibility with 3D image analysis, aimed to identify ducts within the intact mammary stroma and to subsequently recognize all ductal EYFP⁺ cells. Ductal EYFP⁺ cells were classified as luminal or basal based on the proportion of K8 versus SMA fluorescence signal. For a volumetric analysis, the volume ratio

of EYFP⁺ cells within each duct was computed with respect to the entire ductal (cellular) volume, and the intensity of K8 in EYFP⁺ cells was also compared with the overall K8 intensity level within the duct. For computational efficiency, a multi-resolution transform[38] was used for the K8 channel; a coarse scale was used to detect the duct and the full detail scale was used to identify voxels significantly different from the background. The coarse scale was segmented with a robust threshold, obtained as the median of the intensity values in the transformed stack plus three times the median absolute deviation of these values. The up-sampled structure represents an approximation of the duct (Supplementary Fig. 14b) and the sum of all its voxels was a measure of the volume of the duct. Within the duct, significant voxels (excluding intercellular spaces and nuclei) were detected plane-wise from the fine detail levels of the wavelet coefficients of the two-dimensional wavelet transformed image by applying a false discovery rate-based thresholding[39]. Independently, EYFP⁺ cells were identified after a difference of Gaussian filtering suited to the noise level of the image; as the filter was applied on the full-resolution 3D stack, a recursive filter implementation in CImg (Deriche, CImg) was used for time efficiency. The threshold was computed as above, as the robustly estimated 99% quantile of the Gaussian distribution of filtered intensity values: the median plus three times the median absolute deviation of these intensities (Supplementary Fig. 14c). Subsequently, only EYFP⁺ cells inside the detected duct were taken into account. These cells were classified as luminal or basal based on the comparison of intensity values in the K8 and SMA channels of the voxels belonging to each segmented cell; for a chosen threshold, the voxels exceeding this threshold in the K8 and SMA channels, respectively, were counted. If the number of K8 voxels exceeded the SMA, the cell was classified as luminal, otherwise it was classified as basal (Supplementary Fig. 14d). Note that a perfect exclusion of one colour cannot be expected due to the resolution limits of the optical system. To provide robustness with respect to

the choice of the intensity threshold, the classification was performed using a multi-threshold approach, with levels 100, 300, 500, up to 1,500 and the majority vote for all thresholds gave the final classification of the cell. Finally, a Kolmogorov–Smirnov test was performed to determine if the significant voxels of the duct in K8 channel were differentially distributed compared with the significant voxels inside the segmented $EYFP^+$ cell.

Some particularities of the basal $EYFP^+$ co-localization images (related to clone YP.3) such as the poorer signal-to-noise ratio of these images (due to the depth of this clone within the mammary fat pad) and the elongated shape of the $EYFP^+$ cells, made a few modifications of the described analysis necessary. To improve the quality of the images, a denoising step was applied followed by a fast deblurring step (Dr Jerome Boulanger, Medical Research Council—Laboratory of Molecular Biology, private communication). The segmentations are performed in 3D for all channels, however, to separate elongated and overlapping $EYFP^+$ cells, a seeded watershed was used (the seeds are thresholded distance images of the inverted segmented $EYFP^+$ image, where the threshold is manually selected). The classification of the $EYFP^+$ cells is performed as before: if the number of K8 voxels exceeded the SMA, the cell was classified as luminal, otherwise it was classified as basal. Using this analysis, two cells were excluded from the classification due to their localization in regions where the SMA signal was undetectable and thus the double/nested tubular structure could not be observed.

Quantification of $PR^+$ and $K8^{hi}$ cells (Supplementary Fig. 3b) was performed on maximum intensity projections of 3D image stacks using the Cell Counter plugin in Image J (v1.50a, National Institutes of Health). Maximum intensity projections of PR and K8 channels were scored independently. At least 200 cells were counted per image, with six images analysed from three independent mice (total cells counted: 1,831). Manual counting of $EYFP^+$ cells (Fig. 3d and Supplementary Fig. 10b) was performed on 10 histological sections cut $>25\,\mu m$ apart, and spanning $\sim 300\,\mu m$. K8 was used to mark the luminal lineage and SMA was used to mark basal cells.

The number of alveoli that were fully or partially populated by $EYFP^+$ cells of a single lineage were manually counted in Image J.

**Statistics.** The Kolmogorov–Smirnov test (Supplementary Fig. 10a) was performed in MATLAB (R2014a, The MathWorks Inc., Natick, Massachusetts). All values are shown as mean ± s.d.

**Data availability.** The data supporting the findings of this study are available within the article and its Supplementary Information files. All other relevant source data are available from the authors on request.

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

## Acknowledgements

We thank the Department of Pathology Biological Services Unit for help with animal work, H. Skelton for help with histology and Peter Humphreys and Dr Martin Lenz for imaging support. This work was supported by a grant from the Medical Research Council programme grant no. MR/J001023/1 (B.L-L. and C.J.W). F.M.D. was funded by a National Health and Medical Research Council CJ Martin Biomedical Fellowship (GNT1071074). O.B.H. was funded by a Wellcome Trust PhD studentship (105377/Z/14/Z).

## Author contributions

B.L-L., F.M.D. and O.B.H. performed all experiments; B.L.-L., F.M.D., O.B.H., S.K., D.J.W. and C.J.W. conceived and designed the experiments; L.M. designed and performed the 3D image analysis; B.L.-L., F.M.D and C.J.W. wrote the manuscript.

## Additional information

**Competing financial interests:** The authors declare no competing financial interests.

