## [Peer Review File · Nature Communications]

Reviewer #1 (Remarks to the Author)

The authors have adequately addressed all of my remaining questions.

Reviewer #4 (Remarks to the Author)

In this study, Davis et al. use a novel approach to define the hierarchy within bilayered mammary epithelium. The results of this work suggest that luminal and basal lineages are maintained by their respective unipotent stem cells. Another interesting conclusion is that the progeny of a single stem cell can be dispersed in the mammary epithelium. The "slippage" technique used by the authors permits initial labelling of few cells only, whereas the 3D-imaging technique permits an unbiased analysis of the labelled progeny. The clones developed can be visualized in 3D, so that the entire clone can be examined in each case. In contrast to previous studies, the labelling of the cell to be traced does not depend on the activity of any specific promoter and can happen in any cell that divides.

The study is very important, it deals with a fundamental aspect of mammary gland biology - the stem cell properties. The technology used is novel and pertinent, i.e., the one that helps to resolve currently existing contradictions regarding mammary stem cell functions.

The authors have adequately addressed the criticism of previous reviewers.

In particular, the authors have addressed my queries, they show that some labelled cells can survive several pregnancies. However, these cells do not seem to be very numerous, and the text has been corrected accordingly.